# DEN-Induced Rat Model Reproduces Key Features of Human Hepatocellular Carcinoma

**DOI:** 10.3390/cancers13194981

**Published:** 2021-10-04

**Authors:** Keerthi Kurma, Olivier Manches, Florent Chuffart, Nathalie Sturm, Khaldoun Gharzeddine, Jianhui Zhang, Marion Mercey-Ressejac, Sophie Rousseaux, Arnaud Millet, Herve Lerat, Patrice N. Marche, Zuzana Macek Jilkova, Thomas Decaens

**Affiliations:** 1Université Grenoble Alpes, 38000 Grenoble, France; keerthi.kurma@chu-montpellier.fr (K.K.); florent.chuffart@univ-grenoble-alpes.fr (F.C.); kgharzeddine@chu-grenoble.fr (K.G.); zhangjh20@fudan.edu.cn (J.Z.); mressejac@chu-grenoble.fr (M.M.-R.); sophie.rousseaux@univ-grenoble-alpes.fr (S.R.); arnaud.millet@inserm.fr (A.M.); herve.lerat@univ-grenoble-alpes.fr (H.L.); patrice.marche@inserm.fr (P.N.M.); 2Institute for Advanced Biosciences, Research Center Inserm U 1209/CNRS 5309, 38700 La Tronche, France; Olivier.Manches@efs.sante.fr; 3Etablissement Français du Sang, Rhone-Alpes Auvergne, 38043 Grenoble, France; 4Service d’Anatomo-Pathologie, Pôle de Biologie, CHU Grenoble Alpes, 38700 La Tronche, France; NSturm@chu-grenoble.fr; 5Research Department, CHU Grenoble Alpes, 38700 La Tronche, France; 6Service d’Hépato-Gastroentérologie, Pôle Digidune, CHU Grenoble Alpes, 38700 La Tronche, France; 7Unité Mixte de Service Université Grenoble Alpes hTAG, Inserm U046, CNRS UAR2019, 38700 La Tronche, France

**Keywords:** DEN-induced rat model, HCC, liver cancer, hepatocarcinogenesis

## Abstract

**Simple Summary:**

Hepatocellular carcinoma is the most frequent form of primary liver cancer, characterized by increasing incidence and high mortality. Animal models of hepatocellular carcinoma are widely used to study the biology of cancer and to test potential therapies. Herein, we describe how the rat model of DEN-induced hepatocellular carcinoma mimics the pathogenesis of hepatocellular carcinoma seen in humans, including liver damage, chronic inflammation, hepatocytes proliferation, liver fibrosis and cirrhosis, disorganized vasculature, and modulations of the liver’s immune microenvironment. Our results should help the hepatocellular carcinoma field to better tailor the use of the DEN-induced rat liver cancer model for testing specific experimental hypotheses or to perform preclinical testing.

**Abstract:**

Hepatocellular carcinoma (HCC) is the most common type of liver cancer. The majority of HCC cases are associated with liver fibrosis or cirrhosis developing from chronic liver injuries. The immune system of the liver contributes to the severity of tissue damage, the establishment of fibrosis and the disease’s progression towards HCC. Herein, we provide a detailed characterization of the DEN-induced HCC rat model during fibrosis progression and HCC development with a special focus on the liver’s inflammatory microenvironment. Fischer 344 male rats were treated weekly for 14 weeks with intra-peritoneal injections of 50 mg/kg DEN. The rats were sacrificed before starting DEN-injections at 0 weeks, after 8 weeks, 14 weeks and 20 weeks after the start of DEN-injections. We performed histopathological, immunohistochemical, RT-qPCR, RNA-seq and flow cytometry analysis. Data were compared between tumor and non-tumor samples from the DEN-treated versus untreated rats, as well as versus human HCCs. Chronic DEN injections lead to liver damage, hepatocytes proliferation, liver fibrosis and cirrhosis, disorganized vasculature, and a modulated immune microenvironment that mimics the usual events observed during human HCC development. The RNA-seq results showed that DEN-induced liver tumors in the rat model shared remarkable molecular characteristics with human HCC, especially with HCC associated with high proliferation. In conclusion, our study provides detailed insight into hepatocarcinogenesis in a commonly used model of HCC, facilitating the future use of this model for preclinical testing.

## 1. Introduction

Liver cancer is currently the third most common cause of cancer-related death worldwide [1], with hepatocellular carcinoma (HCC) accounting for the majority of these cases. Almost all HCC cases are associated with liver fibrosis or cirrhosis developed from chronic liver injuries. Although each underlying condition might involve different carcinogenic pathways, fibrosis and cirrhosis are regarded as crucial factors in the carcinogenesis of human liver tissue. The immune system of the liver contributes to the severity of the necrotic-inflammatory tissue damages, the establishment of fibrosis and the disease’s progression towards HCC [2]. Since fibrotic liver is characterized by modified liver vascularization, extracellular matrix composition and drug metabolism, fibrotic animal models are highly relevant to test HCC drugs for their efficacy against tumor initiation and/or progression [3,4]. The diethyl nitrosamine (DEN) chronically induced rat model reproduces human liver fibrosis and cirrhosis leading to HCC development [5,6], but limited information exists about pathway alterations, the involvement of inflammation, or the immune system characteristics of this model during progression to carcinoma. In this study, we thoroughly characterized DEN-induced HCC rat models during the progression and development of fibrosis and HCC, with a special focus on tumor immune microenvironments. 

Our results show that in the DEN-induced HCC rat model, the process of hepatocarcinogenesis includes liver damage, hepatocytes proliferation, liver fibrosis/cirrhosis, disorganized vasculature, chronic inflammation and modulations of the liver’s immune microenvironment, which recapitulate the main events observed in human HCC. Overall, our findings comprehensively characterize the DEN-induced HCC rat model, and demonstrate that it largely mimics the pathological process of human HCC, including features of the tumor microenvironment. In this context, the DEN-induced cirrhotic HCC rat model is a relevant pre-clinical tool to evaluate the new HCC treatment’s efficacy and tolerance in a liver cirrhotic background.

## 2. Materials and Methods

### 2.1. Rat Model 

Seven-week-old Fischer 344 male rats (Charles River Laboratories, Écully, France) were treated weekly with intra-peritoneal injections of 50 mg/kg DEN (Sigma-Aldrich, Steinheim am Albuch, Germany). The injections were performed once a week and diluted in pure olive oil in order to obtain HCC on a fibrotic/cirrhotic liver after 14 weeks. We analysed four different time points of DEN treatment: the 0-week group was sacrificed before the start of any DEN injections, the 8-week group after 8 weeks of DEN injections, the 14-week group after 14 weeks of DEN injections and the 20-week group after 14 weeks of DEN injections followed by 6 weeks without DEN injections. Control animals of the same age, treated with no DEN, were used for the 8 w, 14 w and 20 w time points. The animals were housed at Plateforme de Haute Technologie Animale (PHTA) at the University of Grenoble-Alpes core facility (Grenoble, France), EU0197, License no. #C38-51610006 from Direction Départementale de la Protection des Populations., under specific pathogen–free conditions, a temperature-controlled environment with a 12 h light/dark cycle and ad libitum access to water and diet. The animal housing and procedures were conducted in accordance with the recommendations from the Direction des Services Vétérinaires, Ministry of Agriculture of France, according to the European Community Council Directive 2010/63/EU and according to recommendations for health monitoring from the Federation of European Laboratory Animal Science Associations. The protocols involving animals were reviewed by the local ethics committee, “Comité d’Ethique pour l’Expérimentation Animale no.#12, Cometh-Grenoble”, and approved by the French Ministry of Research (#12900-2018010416469902 v4).

### 2.2. Immunohistochemical and Immunofluorescence Analyses 

The liver tissues were fixed in formalin solution, neutral buffered at 10% (Sigma-Aldrich, Steinheim am Albuch, Germany) and paraffin-embedded; four-micrometer sections of tissue were prepared. Hematoxylin-eosin (HE) staining was used for the histopathological examination. The grading of inflammatory activity and staging of fibrosis were performed according to the METAVIR scoring system, a histological scale used to quantify the degree of fibrosis (F). “F” refers to the extent of fibrosis and varies from F0 to F4 (F0 = no fibrosis, F1 = portal fibrosis without septa, F2 = portal fibrosis with rare septa, F3 = numerous septa without cirrhosis, and F4 = cirrhosis). The development of fibrosis and HCC was determined by an experienced pathologist who was blind to the study groups.

To detect HCC development and cancer stem cells in the liver tissues, the paraffin-embedded sections were incubated overnight at 4°C with the primary anti-GST-P (rabbit pAb, MBL International, Woburn, MA, USA) or anti-CD133 (Gene Tex, Irvine, CA, USA). To detect proliferating cells, the sections were incubated overnight at 4 °C with the primary anti-Ki67 antibody (clone SP6, Thermofisher scientific, Rockford, IL, USA) or with the anti-Cyclin D1 antibody (Abcam, clone EPR2241, Cambridge, UK). To analyze macrophages, the sections were incubated overnight at 4 °C with primary anti-mouse CD68/SR-D1 (Novusbio-NB600-985, clone ED1). The anti-rabbit and anti-mouse EnVision system HRP Labelled Polymer (Dako Agilent, Santa Clara, CA, USA) was followed by 3,3′-Diaminobenzidine (DAB) for immune detection. Collagen was detected on the paraffin-embedded sections with picro-sirius red stain solution (Sigma-Aldrich, Steinheim am Albuch, Germany). The positive area was quantified using ImageJ software (NIH, Bethesda, MD, USA) on 10–15 randomly selected fields/sections captured by an Olympus BX41 microscope. All the analyses were performed in a double-blinded manner.

To detect vascularization, the paraffin-embedded sections were incubated with an anti-rat CD34 antibody (goat pAb, R&D systems, Minneapolis, MN, USA) followed by incubation with Alexa 647-conjugated donkey anti-goat IgG (Life Technologies, Carlsbad, CA, USA). Images were captured using the ApoTome microscope (Carl Zeiss AG, Oberkochen, Germany) equipped with an AxioCam MRm camera and collected by AxioVision software. The positive area threshold was quantified using ImageJ software (NIH, Bethesda, MD, USA) on 10 randomly selected fields/sections (10× magnification). All the analyses were performed in a double-blinded manner.

### 2.3. Real-Time Polymerase Chain Reaction (PCR) 

The total RNA was extracted from rat liver tissue samples preserved with an RNA stabilization solution (Thermo scientific, Rockford, IL, USA). The RNA purification was performed with RNeasy Mini Kit^®^ (Qiagen, Hilden, Germany). The reverse transcription was performed with an iScriptTM reverse transcription supermix Kit (BioRad, Hercules, CA, USA), and the amplification reactions were performed in a total volume of 20 µL by using a Thermocycler sequence detector (BioRad CFX96, Hercules, CA, USA) with an iTaqTM Universal SYBR^®^Green super mix qPCR kit (BioRad, Hercules, CA, USA). GADPH was used as a housekeeping gene. The primers listed in Appendix A were designed with Primer 3 software (version 4.0.0, Michelstadt, Germany) and verified on BLAST. The oligonucleotide sequences were synthesized by Eurofins Genomics^®^ in a 0.01 µmol scale, with a salt-free level of purification. Every analysis was performed in duplicates.

### 2.4. Flow Cytometry Analysis

The liver tissue samples collected in RPMI media were resolved into a single cell suspension by mechanical disruption. Later, the cells were washed with PBS (1×) and stained for multi-parametric flow cytometry analyses. A Zombie UV™ Fixable Viability Kit (BioLegend, San Diego, CA, USA) was used to detect nonviable cells. The cells were immunostained without any stimulation, with the following extracellular anti-rat antibodies: CD45, CD3, CD8, CD4 and CD25. The cells were then permeabilized and fixed with a Foxp3/Transcription kit (Thermo Fisher scientific, Rockford, IL, USA) and stained with the intracellular anti-rat Foxp3 antibody. Isotype-matched antibodies were used as controls. The data were on a BD-LSRII flow cytometer with BD FACSDiva 6.3.1 software and analyzed using FCS Express 6 PLUS software (version, Pasadena, CA, USA).

### 2.5. Enzyme-Linked Immunosorbent Assay (ELISA) Analysis

The TNF-alpha DuoSet ELISA Kit (R&D Systems, Minneapolis, MN, USA), IFN-gamma Uncoated ELISA (Thermo Fisher Scientific, Rockford, IL, USA), IL-4 Ready-Set-Go Elisa kit (E030212, eBiosciences, Thermo Fisher Scientific, Rockford, IL, USA) and IL-10 BD OptEIA Set (8029217EU) were used as recommended by providers. 

### 2.6. RNA-Seq

The total RNA was extracted from rat liver tissue samples preserved with an RNA stabilization solution (Thermo scientific, Rockford, IL, USA). The RNA purification was performed with RNeasy Mini Kit^®^ (Qiagen, Hilden, Germany), according to the instructions provided by the manufacturer; and quality assessment for the RNA Integrity Number (RIN) was performed using Agilent 2100 Bioanalyzer (Agilent, Palo Alto, CA, USA). The libraries were prepared using the Illumina TruSeq Stranded mRNA kit (Illumina, San Diego, CA, USA), according to the manufacturer’s protocol, starting with 1 μg total RNA. The mRNAs were purified, fragmented and converted to cDNA with reverse transcriptase. The resulting cDNAs were converted to double-stranded cDNAs and subjected to end-repair, A-tailing, and adapter ligation. The constructed libraries were amplified using 15 cycles of PCR. The libraries were quantified by qPCR with the LightCycler 480 real-time PCR instrument (Roche, Basel, Switzerland). The normalized and pooled libraries were sequenced on a NovaSeq 6000 platform with S1 flow cells (200 cycles) (Illumina, San Diego, CA, USA) in a 75 bp paired-end.

The fastq files were aligned on the UCSC rn6 genome using STAR (2.7.1a) [7] to produce bam files. The bam files were counted using the HTSeq framework (0.11.2) [8] (with the following options: -t exon -f bam -r pos–stranded = reverse -m intersectiostrict –nonunique none). The normalization and differential analysis were performed using the R software (R Core Team. R: A language and environment for statistical computing. Vienna, Austria: R Foundation for Statistical Computing, 2017), DESeq2 (1.22.2) [9,10] and SARTools [11] packages. 

The raw sequencing data are available at NCBI’s Gene Expression Omnibus under the GEO Series accession number GSE182860 (https://www.ncbi.nlm.nih.gov/geo/query/acc.cgi?acc=GSE182860, accessed on 31 December 2020).

The GSEA was conducted with the pre-ranked GSEA method [12] within the KEGG, and Hallmark databases (https://broadinstitute.org/msigdb, accessed on 31 December 2020). The single-sample GSEA scores, representing the degree to which the genes in a particular gene set are coordinately up- or down-regulated within a sample, were calculated using the GSVA program [13], following the methodology described previously [14]. A volcano plot displaying differential gene expression analysis of RNA-seq data comparing 8 w DEN vs. 8 w No DEN or 14 w DEN non-tumoral vs. 14 w No DEN used a significancy threshold for a relative expression fold change ≤ −2.0 or ≥ 2.0 and adjusted *p* ≤ 0.01. Volcano plots displaying the differential gene expression of the top 25 over-expressed and top 25 under-expressed genes in human HCC used a significancy threshold ≤ −0.5 or ≥ 0.5 for a relative expression fold change and adjusted *p* ≤ 0.05. 

### 2.7. Statistical Analysis

All the data were tested for normality and the appropriate statistical test was chosen. The comparisons of means were calculated by using ANOVA tests with Tukey HSD correction for multiple means comparisons, and independent *T*-tests only when two means were compared. The data are presented as mean values ± standard error mean (SEM). The statistical analyses were performed using Prism 6 (GraphPad Software Inc., San Diego, CA, USA).

## 3. Results

### 3.1. Chronic DEN Promotes Hepatocarcinogenesis Associated with High Cell Proliferation, Fibrosis and Abnormal Vasculature

To characterize the hepatocarcinogenesis induced by DEN, Fisher 344 rats were injected weekly with intra-peritoneal injections of DEN (50 mg/kg) for 14 weeks (Figure 1a), which caused progressive liver damage (Appendix A) and hepatocarcinogenesis, leading to nodule development in 100% of the animals after 14 weeks of injections (Figure 1b,c). GST-P^+^ preneoplastic lesions were already visible after 8 weeks of DEN injections (*p* < 0.0001) and strongly expanded at 14 weeks (Figure 1d and Appendix A). This transformation of early lesions was accompanied by an increase in CD133^+^ stem cells (*p* < 0.0001) (Figure 1e). Consistently, DEN treatment strongly promoted hepatocyte proliferation, as assessed by Ki67 and cyclin D1 staining (Figure 1f).

We next analyzed fibrosis development using Sirius red staining, which showed a significant increase in fibrosis at 8 weeks (*p* = 0.0009) and 14 weeks (*p* < 0.0001), compared to 0 weeks (ANOVA, *p* < 0.0001), (Figure 2a). The METAVIR score analyses confirmed that the severity of the fibrosis increased from no fibrosis at 0 weeks to portal fibrosis without and with septa at 8 weeks, leading to severe fibrosis or cirrhosis in 55% of the animals at 14 weeks (Appendix A). The qPCR analyses of Collagen-1, alpha smooth muscle actin (α-SMA), transforming growth factor beta (TGF-β) and a tissue inhibitor of metalloproteinases (TIMP-1) confirmed the development and progression of liver fibrosis/cirrhosis as a result chronic DEN treatment, which was associated with a non-significant decrease in matrix metalloproteinase (MMP)2 and MMP9 expression (Figure 2b).

Human liver cirrhosis is generally associated with distortion of the hepatic architecture and disorganized vasculature [15]. Indeed, we found that DEN treatment induced modifications leading to abnormal vasculature with a significant increase in the CD34 positive area (Figure 2c) at 14 weeks of injections. The molecular mechanisms responsible for DEN-induced liver modifications were further investigated by RNA-sequencing (RNA-seq) comparing DEN-treated animals with untreated (no DEN) animals of the same age. Gene set enrichment analysis (GSEA) revealed that genes upregulated following a DEN treatment were significantly enriched in genes involved in the cell cycle division and proliferation, with G2M checkpoint and E2F targets being the top gene sets after 8 and 14 weeks (Figure 2d,e, respectively), including, for example, Tacc3, Stmn1, Spc25 and Slc7a1 (Figure 2f,g). Similarly, DEN treatment was associated with the enrichment of a gene set related to epithelial-mesenchymal transition (EMT), which includes Lama5, Flna, Fas, Col5a2, Col4a1, Col16a1, Dcn, Sgcb, Cdkn1, Mybl2, and Fbn1 (Figure 2d–g). 

In summary, chronic DEN injections lead to liver damage, hepatocytes proliferation, liver fibrosis/cirrhosis and disorganized vasculature, which mimics the usual events observed during human HCC development.

### 3.2. DEN-Induced Rat Model of HCC Shares Genetic and Molecular Characteristics with Human HCC

To characterize the mechanism of irreversible malignant transformation induced by DEN, we next investigated tumor nodules. For this purpose, the DEN injections were stopped after 14 weeks and the animals were left untreated for 6 weeks (20 w group, Figure 1a). We observed no significant difference in the incidence of tumors between the 14 weeks and the 20 weeks group but the size of the tumors increased from 2.65 ± 0.22 mm at 14 weeks to 4.89 ± 0.53 mm at 20 weeks, *p* = 0.0003 (Figure 3a, Appendix A). The nodules were histologically confirmed as HCC (Figure 3b) and were characterized by high hepatocytes proliferation and disorganized vasculature (Appendix A).

We then looked for similarities in the gene expression patterns between DEN-induced tumors in rats and human HCC tumors. For this purpose, the top 25 over-expressed and bottom 25 under-expressed genes in human HCC were identified using available HCC TCGA RNAseq data, (Appendix A) [16]. We found that DEN-induced HCC in our rat model over-expressed 21 genes out of the top 25 genes over-expressed in human HCC, including genes whose high expression is associated with unfavorable prognosis in liver cancer, such as Spp1, Ube2c, Pttg1, Ube2t, Ccnb1, Nt5dc2, G6pd, Cenpw, Tk1, Tacc3 and Stmn1 (Figure 3c, Appendix A). Among the bottom 25 under-expressed genes in human HCC we also found genes under-expressed in the tumors of the DEN-induced rat model, which included Cxcl14, Pzp, Cyp1a2, Bco2, Vipr1, Pth1r (Figure 3c). A GSEA approach using gene sets available in the MsigDB (https://www.gsea-msigdb.org/gsea/msigdb/, accessed on 31 December 2020) showed that genes overexpressed in the tumors of rats treated by DEN were significantly enriched in genes upregulated in a subclass of human HCC, with increased proliferation and chromosomal instability [17] (Figure 3d). Moreover, a GSEA comparing tumor versus non-tumor tissues in DEN-treated rats in the 20 weeks group identified a significant enrichment in gene sets related to E2F targets, EMT, G2M checkpoint, Myc targets and KRAS signaling (Appendix A), while the most significantly depleted gene sets were related to bile acid metabolism, xenobiotic metabolism, fatty acid metabolism, peroxisome and oxidative phosphorylation (Appendix A). A single-sample GSEA of our rat model using gene sets classically associated with the development of human HCC demonstrated that the angiogenesis, p53 pathway, IL2 STAT5 signaling and IL6 JAK STAT3 signaling gene sets were consistently upregulated following DEN treatment (Figure 3e). Other gene sets, such as Hedgehog signaling, MTOR1 signaling, PI3K AKT MTOR signaling, VEGF signaling or Notch signaling, showed a high heterogeneity in their enrichment scores in different tumors, which is also typical of human liver cancer [18,19,20]. Altogether these results suggest that DEN-induced liver tumors in our rat model share remarkable genetic and molecular characteristics with human HCC, especially with HCC associated with high proliferation.

### 3.3. DEN-Induced Modulations of the Tumor Immune Microenvironment

The immune system has been shown to play crucial roles in the development of liver cancer [21]. To detail the DEN-induced modulations of the inflammatory microenvironment in the liver that contribute to HCC development, we used an immune-mediated cancer field (ICF) gene expression signature associated with the development of human HCC. ICF molecular subclass was stratified based on lymphocyte infiltration and on the activation of either immunosuppressive or pro-inflammatory signals, as presented previously [22]. We observed that DEN treatment was strongly associated with the high-infiltrate ICF signature after 8 weeks of injections (Figure 4a), which was confirmed by the upregulation of CD4 and CD8 mRNA at 8 weeks, compared to 0 weeks (Figure 4b). 

Similarly, a GSEA of RNA-seq data of hepatic tissue from DEN-treated rats showed enrichment for genes involved in TNF signaling via NF-κB and inflammatory responses (Appendix A). The relative expression of genes based on the Danaher inflammation signature, a robust gene signature for tumor-infiltrating immune cells [23,24], indicated that intrahepatic T cells and macrophages were promoted by 8 weeks of DEN-injections (Appendix A). Similarly, immuno-histochemistry corroborated these data, showing an increased CD68^+^ cell accumulation in the liver already detectable after 8 weeks of DEN treatment (Figure 4c). 

The HCC nodules were strongly associated with the pro-tumorigenic immunosuppressive ICF subclass at 14 weeks (Figure 4a) and, accordingly, the flow cytometry analysis of the intrahepatic cells revealed a lower frequency of CD8^+^ T cells in tumoral compared to non-tumoral tissue (Figure 4d), while the frequency of T_reg_ per CD4^+^ T cells was increased (Figure 4e). The protein-based analyses of IFN-γ and TNF-α confirmed these findings, showing significantly lower levels of pro-inflammatory cytokines in tumoral parts compared to non-tumoral parts (Figure 4f). Next, we focused on the immune checkpoint molecules that are known to be expressed and involved in human HCC. In our rat model, we observed that genes coding for immune checkpoint molecules, such as lgals9 (Galectin 9), cd44, cd48, cd276 (B7H3), or tnfrsf9 (4-1BB), were modulated by DEN injections (Appendix A), suggesting their involvement in tumor microenvironment modulations. 

To conclude, we observed an extensive tumor immune microenvironment heterogeneity in the tumor nodules of DEN-induced rats (Figure 4a, Appendix A) which is one of the typical hallmarks of tumor evolution and progression in human HCC [25].

## 4. Discussion

Newly developed drugs for HCC should be pre-clinically tested in an appropriate animal model that is well characterized and mimics major oncogenic events and tumor microenvironment modulations observed in human hepatocarcinogenesis. HCC animal modelling has historically relied on the use of a variety of carcinogens (such as DEN) that efficiently induce liver tumors in mice or rats. However, the detailed characteristics of the tumor and tumor microenvironment of these models often remain unknown, and therefore their validity in faithfully modelling human HCC is unclear. 

A classical mouse model of DEN-induced HCC uses a single injection with DEN at postnatal day 15, which later leads to the formation of a tumor after 25 weeks [26], but does not induce chronic inflammation or liver fibrosis. Since human HCC is almost universally linked with chronic inflammation and fibrosis, it is not surprising that these DEN-induced mouse models result in tumors that are distinct from human HCC [26,27], although recent analyses suggest similarities between the mouse DEN-induced HCC and alcohol-induced HCC in patients [28]. To overcome the absence of fibrosis, a classical mouse DEN-induced HCC needs to be combined with the repeated injection of a pro-fibrogenic agent CCl4 [29] to simulate the features of human liver fibrosis. 

By contrast, a classical rat model of DEN-induced HCC is based on chronic exposure to DEN which, by itself, leads to chronic inflammation and fibrosis, followed by HCC after 14–20 weeks [5,6,30]. However, while the DEN-induced mouse model of HCC is well described and characterized, there is still limited information about the DEN-induced rat model, despite this model being widely used in preclinical testing to target liver fibrosis and HCC [5,6,30,31,32,33]. Here, we performed a characterization, including an RNA-seq analysis, to identify the molecular signatures associated with DEN-induced liver cancer development in rats and compared them with the expression profiles of human HCC. We observed that the development of DEN-induced HCC in the rat is associated with liver damage, hepatocytes proliferation, liver fibrosis/cirrhosis, disorganized vasculature and chronic inflammation; all these patterns are also associated with human HCC. 

Importantly, the gene expression patterns of DEN-induced HCC in the rat were found to be similar to human HCC expression profiles. Most of the top 25 genes overexpressed in human HCC were also upregulated in DEN-induced tumors in rats. Notably, GSEA also revealed that DEN-induced tumors in rats are strongly enriched in genes associated with a human HCC “high proliferation” subtype [17]. In addition, we observed a significant enrichment in the gene set related to Myc targets in tumor tissue compared to nontumoral tissue, while the TGF beta-Wnt pathway was not affected in this rat model, which recalls the previously proposed proliferative subtype S2 [34,35]. 

A key step in human HCC progression is EMT [36], a complex process that allows epithelial cells to gain mesenchymal features, including properties of migration and invasiveness. Furthermore, EMT was also one of the key processes involved in DEN-induced hepatocarcinogenesis in rats, suggesting that this model can be used to preclinically test new treatments targeting EMT. 

Human HCC is a very heterogeneous disease and tumor heterogeneity seems to be even more complex in multifocal HCC patients than in patients with single tumor in the liver [20]. The most frequently altered pathways in human HCC involve growth factor receptors, such as VEGFR, EGFR, FGFR, IGFR, and their intermediates, including PI3K -AKT-MTOR. Other key altered pathways are related to cell differentiation, including Wnt/β-catenin, JAK/STAT, Hedgehog or Notch signaling pathways. Notably, we observed a high heterogeneity in the pathway alterations involved in the development of DEN-induced tumors in rats, which is typical for HCC models that are developed based on the induction of chronic liver inflammation [3,27]. Moreover, the formation of neoplastic lesions and dysplastic nodules may have contributed to the heterogeneity of the molecular findings. The mutational landscape of chronic DEN-induced HCC in Sprague-Dawley rats was recently presented, showing heterogeneous mutation signatures matched with the previously identified signatures for human liver cancers [37]. 

HCC tumors are complex ecosystems also incorporating non-tumoral cells. Today it is clearly established that the tumor micro-environment, in particular the dysfunctional immune system, largely contributes to HCC development. Humanized rodents would represent one of the most ideal preclinical models for the testing of therapies targeting the tumor-immune system interface [3]. However, they are not yet well established in the HCC field and it is therefore necessary to evaluate whether currently used models recapitulate the tumor-immune system interactions observed in human HCC. Our results show that the immune system is strongly modified by DEN injections, shifting from a high immune infiltrate profile at the beginning to a more immunosuppressive profile at the time of HCC occurrence, which probably further contributes to the progression of HCC. Indeed, the tumor nodules were characterized by diminished levels of inflammatory cytokines IFNγ and TNFα when compared to non-tumoral tissue, while the intratumoral frequency of T_reg_ was enhanced, which is classically associated with HCC progression and poor survival [21,38,39,40]. Interestingly, the gene expression of several immune checkpoint molecules in the DEN-induced tumor microenvironment is clearly different from the immune checkpoint landscape associated with human HCC. Other immune checkpoint molecules seem to be classically expressed at the gene level, and they are modulated during DEN treatment. For instance, B7-H3, the key inhibitor of tumor antigen-specific immune response, is increased in hepatocarcinogenesis in the DEN-induced rat model, which is of interest since B7-H3 is considered as an attractive future target for antibody-based immunotherapy [41]. 

Overall, our results show that the model of DEN-induced HCC in rats largely mimics the process of human hepatocarcinogenesis. Still, the main limitation is the rat immune system, which is partially distinct from the human immune system. Thus, all discrepancies in immune responses, including differences in the immune checkpoint landscape, should be taken into account when using this model for preclinical studies of HCC.

## 5. Conclusions

The model of DEN-induced HCC in rats largely recapitulates the pathogenesis of HCC seen in humans and is a relevant animal model for drug development and preclinical testing in the HCC field. 

In conclusion, we expect that these results will help the HCC field to better tailor the use of the DEN-induced rat liver cancer model for testing specific experimental hypotheses or to perform preclinical testing. 

## Figures and Tables

**Figure 1 cancers-13-04981-f001:**
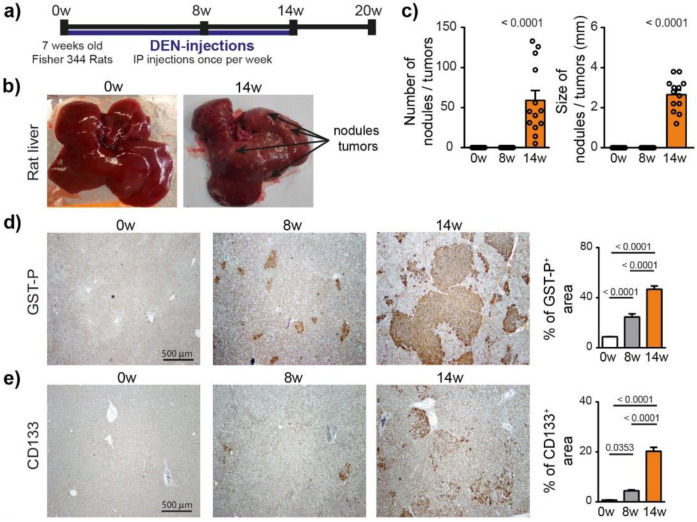
Chronic DEN-induced hepatocarcinogenesis is associated with high hepatocyte proliferation. (**a**) Timeline protocol to characterize DEN-induced cirrhotic rat model of HCC. Seven-week-old Fisher 344 rats were injected weekly by DEN (50 mg/kg per week). The term 0 w indicates the baseline before DEN injections, 8 w indicates 8 weeks of DEN injections, 14 w indicates 14 weeks of DEN injections and 20 w indicates 14 weeks of DEN injections followed by 6 weeks of no DEN. (**b**) Representative pictures of rat liver at 0 w and 14 w. (**c**) Macroscopic examination of livers with assessment of tumor number at the surface of livers and tumor size (average of diameter of the five largest tumors). Each circle represents an individual animal, mean ± SE, *n* = 13/group. (**d**) Representative images of GST-P (4× magnification) and quantification of GST-P+ surface area per high power field (HPF). Mean ± SE, *n* = 9/group. (**e**) Representative images of CD133 staining (4× magnification) and quantification of CD133^+^ surface area per HPF. (**f**) Representative images of nuclear Ki67 and CyclinD1 staining (20× magnification-black arrows) and quantification of nuclear Ki67^+^ and CyclinD1^+^ hepatocytes per HPF. Mean ± SE, *n* = 9/group. The comparison of means was performed using the ANOVA test with Tukey correction, w = weeks.

**Figure 2 cancers-13-04981-f002:**
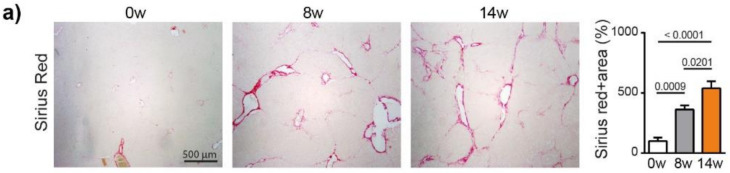
Chronic DEN induces fibrosis and abnormal vasculature. (**a**) Representative images of liver tissue stained with Sirius red (4 × magnification) and quantification of Sirius red-stained area per high power field (HPF) represented on bar graph; 0 w was set as 100%. Values are mean ± SE, *n* = 9/group. (**b**) qPCR analysis of collagen1, α-SMA, TGF-β, TIMP1, MMP2 and MMP9 gene expression in non-tumor liver samples; 0 w was set as 1, values are mean ± SE, *n* = 9/group. (**c**) Representative images of CD34 immunofluorescence staining in liver tissue and quantification of percentage of CD34-stained surface area per HPF (4× magnification). Zero weeks was set as 100%. Values are mean ± SE, *n* = 8–9/group. The comparison of means was performed using the ANOVA test with Tukey correction. The symbol w = weeks. (**d**,**e**) Gene Set Enrichment Analysis (GSEA) of RNA-seq data showing the top 3 gene sets that were significantly enriched after 8 weeks (**d**) or 14 weeks (**e**) of DEN injections compared to the non-treated animals of the same age (8 w no DEN or 14 w no DEN, respectively). NES = normalized enrichment score; FDR = false discovery rate. (**f**,**g**) Volcano plot displaying differential gene expression analysis of RNA-seq data. The log-transformed adjusted *p*-values are plotted on the y-axis and log2 fold change values comparing 8 w DEN vs. 8 w no DEN (**f**) or 14 w DEN non-tumoral vs. 14 w no DEN (**g**) are plotted on the *x*-axis, with significancy threshold for a relative expression fold change ≤ −2.0 or ≥ 2.0 and adjusted *p* ≤ 0.01. The red dots represent only the top upregulated genes related to the three top gene sets. The symbol w = week, NT = non-tumoral.

**Figure 3 cancers-13-04981-f003:**
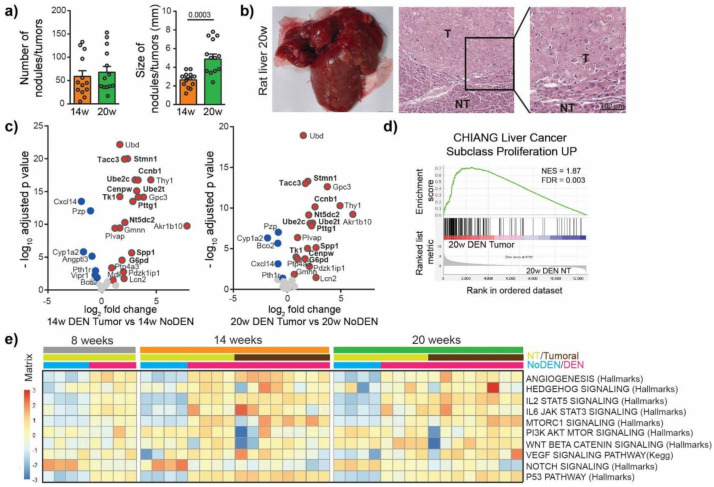
DEN-induced rat model of HCC shows genetic and molecular similarities with human HCC. (**a**) Macroscopic examination of livers with assessment of tumor number at the surface of livers and tumor size (average of diameter of the five largest tumors). Each circle represents an individual animal, mean ± SE, *n* = 13/group. Comparison of means was performed by unpaired *t* test. (**b**) Representative picture of rat liver at 20 w and representative images of hematoxylin and eosin (H&E)-stained sections proving HCC, (20× magnification). T = tumor, NT = Non-tumor tissue. (**c**) Volcano plot displaying differential gene expression in tumors at 14 w (left) and 20 w (right) by RNA-seq. The log-transformed adjusted *p*-values are plotted on the y-axis and log2 fold change values on the *x*-axis, with the threshold for a relative expression fold change ≤ −0.5 or ≥ 0.5 and adjusted *p* ≤ 0.05. Significantly over-expressed genes (red circles) and under-expressed genes (blue circles) in DEN-induced tumors based on top 25 over-expressed and top 25 under-expressed genes in human HCC. Only the top 25 over-expressed and top 25 under-expressed genes in human HCC are shown in this volcano plot. Genes whose high expression is associated with unfavorable prognosis in human liver cancer are marked in bold. (**d**) Gene Set Enrichment Analysis (GSEA) of RNA-seq data from 20 w DEN tumor tissue showing the enrichment of genes associated with human liver cancer subclass characterized by increased proliferation. NES = normalized enrichment score; FDR = false discovery rate. w = weeks. (**e**) Heatmap showing the single-sample GSEA scores for gene sets classically associated with the development of human HCC.

**Figure 4 cancers-13-04981-f004:**
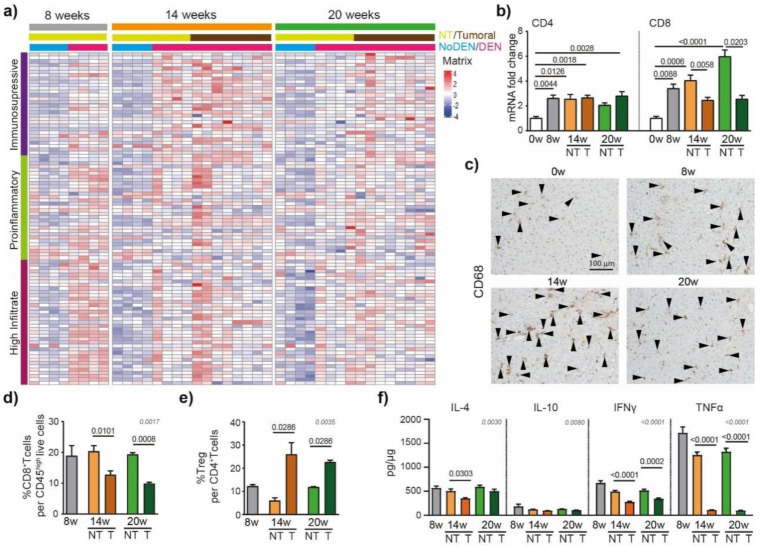
DEN induces modulations of the tumor’s immune microenvironment. (**a**) Heatmap showing the relative expression of genes from the immune-mediated cancer field (ICF) gene expression signature associated with the development of human HCC [22]. Subclasses are stratified based on lymphocyte infiltration and on the activation of either immunosuppressive or pro-inflammatory signals. (**b**) qPCR analysis of CD4 and CD8. The 0 w was set as 1, values are mean ± SE, *n* = 9/group. Comparison of means was performed by ANOVA test with Tukey correction; w = week. (**c**) Representative images of CD68 staining (20× magnification-black arrows). (**d**) Frequency of CD8^+^ T cells per CD45^high^ live cells and (**e**) frequency of Treg (CD25^+^Foxp3^+^) per CD4^+^ T cells in tumoral and non-tumoral liver tissues at 8 w, 14 w and 20 w measured by flow cytometry analysis; values are mean ± SE, *n* = 9/group. (**f**) Interleukin-4 (IL-4), interleukin-10 (IL-10) interferon gamma (INF-γ) and tumor necrosis factor alpha (TNF-α) protein concentration in tumoral and non-tumoral liver tissue at 8 w, 14 w and 20 w measured by Elisa; values are mean ± SE, *n* = 13/group. (**d**–**f**) significant *p*-values (in black) show the comparison between tumoral and non-tumoral values at 14 w and 20 w obtained by Mann–Whitney testing. Comparisons of all the means were performed using the ANOVA test with Tukey correctio; *p*-values in grey.

## Data Availability

The data presented in this study are openly available at NCBI’s Gene Expression Omnibus under the GEO Series accession number GSE182860 (https://www.ncbi.nlm.nih.gov/geo/query/acc.cgi?acc=GSE182860, accessed on 31 December 2020).

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
