# Peer review of "DEN-Induced Rat Model Reproduces Key Features of Human Hepatocellular Carcinoma"

_cancers, 2021, doi:10.3390/cancers13194981_

Round 1

Reviewer 1 Report

In their study Kurma et al. investigated the rat model of DEN-induced hepatocarcinogenesis at different levels using a variety of techniques including histopathology, immunohistochemical stainings, RT-qPCR, RNA-seq, and flow cytometry. With a particular focus on the liver inflammatory microenvironment the authors found that the aforementioned model recapitulates key characteristics of human hepatocarcinogenesis, particularly of those HCCs belonging to the “(high) proliferation class”. The authors conclude that the detailed characterization of the DEN rat HCC model presented in this study will facilitate its optimized use for preclinical testing.

This well written manuscript focusses on a very relevant topic and the results are clearly presented supporting the conclusions drawn.  Thus, I have only minor comments

  • HE stainings corresponding to Figure 1 d) for the 0w, 8w, and 14 time point showing the preneoplastic lesions should be included.
  • In the course of hepatocarcinogenesis one would expect a spectrum of neoplastic lesions. Did the authors also observe dysplastic nodules? If so, is it possible that these may also have contributed to the heterogeneity of the molecular findings or were only "clear cut" HCCs included in the analyses? And (independent of dysplastic nodules) was the heterogeneity of the molecular findings somehow correlated with a histomorphological heterogeneity of the HCCs (e.g. in terms of growth pattern, tumor cell shape, etc)?
  • As discussed in lines 366-374 human HCC is a heterogenous disease with several distinct subtypes. Since the “proliferation class” comprises several previously proposed subtypes (G1-G3, S1 and S2) it would be interesting to point out more detailed how the features of the rat model do or do not mimic these or how they may also show features of the non-proliferation class (e.g IL6 JAK/STAT signaling).
  • There is no information about the criteria that render a gene significantly changed in Figures 2 (f and e) and 3c. This information should appear in the figure legends and the Materials and Methods.

Author Response

Reviewer 1:

In their study Kurma et al. investigated the rat model of DEN-induced hepatocarcinogenesis at different levels using a variety of techniques including histopathology, immunohistochemical stainings, RT-qPCR, RNA-seq, and flow cytometry. With a particular focus on the liver inflammatory microenvironment the authors found that the aforementioned model recapitulates key characteristics of human hepatocarcinogenesis, particularly of those HCCs belonging to the “(high) proliferation class”. The authors conclude that the detailed characterization of the DEN rat HCC model presented in this study will facilitate its optimized use for preclinical testing.

This well written manuscript focusses on a very relevant topic and the results are clearly presented supporting the conclusions drawn.  Thus, I have only minor comments

Response: We thank Reviewer #1 for his/her constructive comments and for appreciating our manuscript.

  • HE stainings corresponding to Figure 1 d) for the 0w, 8w, and 14 time point showing the preneoplastic lesions should be included.

Response: We thank Referee #1 for raising this point. We included HE staining of all groups as Figure S1A.

  • In the course of hepatocarcinogenesis one would expect a spectrum of neoplastic lesions. Did the authors also observe dysplastic nodules? If so, is it possible that these may also have contributed to the heterogeneity of the molecular findings or were only "clear cut" HCCs included in the analyses? And (independent of dysplastic nodules) was the heterogeneity of the molecular findings somehow correlated with a histomorphological heterogeneity of the HCCs (e.g. in terms of growth pattern, tumor cell shape, etc)?

Response: We thank Referee #1 for this comment. Indeed, neoplastic lesions were observed and we used glutathione transferase (GST-P) staining to investigate them. The occurrence of dysplastic nodules was confirmed by the pathologist and it was most significant at 14 weeks when the expression of GST-P was highest.

RNAseq was performed on well-established HCC nodules at 20 weeks. However, we can’t exclude the possibility that the formation of neoplastic lesions and dysplastic nodules was not affecting the RNAseq data at 14 weeks. This information is now discussed in the new version of the manuscript.

Concerning the heterogeneity analyses, we fully agree with the reviewer that studying the correlations or associations between histo-morphological features and molecular characteristics would be very interesting.  Unfortunately, in this study, RNAseq analyses were performed on a subgroup of animals, which was not allowing us to perform correlations with histomorphological characteristics. Thus, this subject needs to be carefully investigated in future studies that should be designed to answer this question.

  • As discussed in lines 366-374 human HCC is a heterogenous disease with several distinct subtypes. Since the “proliferation class” comprises several previously proposed subtypes (G1-G3, S1 and S2) it would be interesting to point out more detailed how the features of the rat model do or do not mimic these or how they may also show features of the non-proliferation class (e.g IL6 JAK/STAT signaling).

Response:  We thank for this excellent suggestion. We detailed this point in the new version of the manuscript including references that are focused on this topic. We observed no modulations of the TGFbeta-Wnt pathway in tumoral tissue of the rat model compared to non-tumoral (or even compared to tissue from animals without DEN injections), while MYC targets signalling was significantly affected. Therefore, we conclude that this model is rather belonging to S2 signature.

Concerning IL6-JAK/Stat3 pathway, we did not observe a significant increase in tumoral part compared to nontumoral part in IL6-JAK/Stat3 signalling. This pathway is rather activated in both non-tumoral part and tumoral part when compared to liver tissue from rats that were not treated by DEN, suggesting the involvement of the pathway in overall liver microenvironment modulation rather than in tumor progression in this rat model.

  • There is no information about the criteria that render a gene significantly changed in Figures 2 (f and e) and 3c. This information should appear in the figure legends and the Materials and Methods.

Response:  We thank Referee #1 for highlighting this inconsistency. This information is now added in the figure legends and the Materials and Methods.

Reviewer 2 Report

Kurma et.al., in their manuscript, "DEN-induced rat model reproduces key features of human hepatocellular carcinoma" characterize the occurrence of Hepatocellular carcinoma in a rat model post injection with diethyl nitrosamine. Their observations suggest that this model recapitulates the hepatocarcinogenesis process similar to human HCC. The typical features of liver damage, hepatocytes proliferation, liver fibrosis/cirrhosis, disorganized vasculature, chronic inflammation and modulations of the liver immune microenvironment were all identical. They capitalized on a few robust test methods of IHC, RNA-seq and FACS analysis to determine and characterize cirrhosis. data is robust, with just a few clarifications:

  1. what is the rationale for choosing the time of 8 weeks, 14 weeks and 20 weeks to sacrifice animals? is there any indication that at these times, there is more clinical manifestation of cirrhosis?
  2. Why was the immunofluorescence and cell cycle analysis limited to 14 weeks? Why was 20 weeks not tested?
  3. Likewise, why were the tumor microenvironmental factors particularly those shown in fig. 4 d-f not tested for 8 weeks? is the microenvironmental perturbation not expected at 8 weeks?

Author Response

Reviewer 2:

Kurma et.al., in their manuscript, "DEN-induced rat model reproduces key features of human hepatocellular carcinoma" characterize the occurrence of Hepatocellular carcinoma in a rat model post injection with diethylnitrosamine. Their observations suggest that this model recapitulates the hepatocarcinogenesis process similar to human HCC. The typical features of liver damage, hepatocytes proliferation, liver fibrosis/cirrhosis, disorganized vasculature, chronic inflammation and modulations of the liver immune microenvironment were all identical. They capitalized on a few robust test methods of IHC, RNA-seq and FACS analysis to determine and characterize cirrhosis. data is robust, with just a few clarifications:

Response: We thank Referee #2 for her/his insights and constructive comments.

  1. what is the rationale for choosing the time of 8 weeks, 14 weeks and 20 weeks to sacrifice animals? is there any indication that at these times, there is more clinical manifestation of cirrhosis?

Response: We thank Referee #2 for this comment. Rats were injected weekly with intra-peritoneal injections of DEN (50 mg/kg) for 14 weeks, which caused progressive liver damage and HCC development leading to nodules development in 100% of animals after 14 weeks of injections. After 14 weeks, DEN injections were stopped. Based on our pilot experiments, including MRI and ultrasound techniques, we choose 8 weeks as a time point before nodules start to appear in animals. Next, we chose 14 weeks as a time point that needs to be characterized as this is the end of DEN injections and we observe that all animals develop nodules on the background of cirrhosis/fibrosis. The last time point, 20 weeks, was selected as the time point "6 weeks after the last DEN injection", and to prevent the loss of animals that is observed after 20 weeks due to tumor bleeding into the peritoneum. 

  1. Why was the immunofluorescence and cell cycle analysis limited to 14 weeks? Why was 20 weeks not tested?

Response: We thank Referee #2 for this question. In the first part of the study, we focused on the effect of DEN injections on the liver, including impact on hepatocytes proliferation, liver fibrosis/cirrhosis, disorganized vasculature etc. As rats were injected for 14 weeks only, we did not present the group 20 weeks in this first part of the manuscript. To characterize the mechanism of irreversible malignant transformation induced by DEN, we next focused on tumor nodules. For this purpose, the DEN injections were stopped after 14 weeks and the animals were left untreated for 6 weeks (20w group).  Group 20 weeks is characterized by disorganized vasculature similarly to week 14 and by significant hepatocytes proliferation. The immunofluorescence and cell cycle analysis of 20 weeks group were now included in the manuscript as Figure S1B.

  1. Likewise, why were the tumor microenvironmental factors particularly those shown in fig. 4 d-f not tested for 8 weeks? is the microenvironmental perturbation not expected at 8 weeks?

Response: We thank Referee #1 for raising this point.  In Figure 4 d-f, we focused on the differences between nontumoral and tumoral tissue. However, as we tested the 8 weeks group as well, we added the results to the new manuscript (new Figure 4) to complete the picture characterizing the microenvironmental perturbation by DEN injections.